# The IL4I1 Enzyme: A New Player in the Immunosuppressive Tumor Microenvironment

**DOI:** 10.3390/cells8070757

**Published:** 2019-07-20

**Authors:** Valérie Molinier-Frenkel, Armelle Prévost-Blondel, Flavia Castellano

**Affiliations:** 1INSERM, U955, Team 09, 94010 Créteil, France; 2Faculty of Medicine, University Paris Est, 94010 Créteil, France; 3AP-HP, H. Mondor - A. Chenevier Hospital, Biological Immunology Service, 94010 Créteil, France; 4INSERM, U1016, Institute Cochin, 75014 Paris, France; 5CNRS, UMR8104, 75014 Paris, France; 6University Paris Descartes, Sorbonne Paris Cité, 75014 Paris, France

**Keywords:** IL4I1, immunosuppression enzyme, tumor, phenylalanine

## Abstract

The high metabolic needs of T lymphocytes in response to activation make them particularly vulnerable to modifications of their biochemical milieu. Immunosuppressive enzymes produced in the tumor microenvironment modify nutrient availability by catabolizing essential or semi-essential amino acids and producing toxic catabolites, thus participating in the local sabotage of the antitumor immune response. L-amino-acid oxidases are FAD-bound enzymes found throughout evolution, from bacteria to mammals, and are often endowed with anti-infectious properties. IL4I1 is a secreted L-phenylalanine oxidase mainly produced by inflammatory antigen-presenting cells—in particular, macrophages present in T helper type 1 granulomas and in various types of tumors. In the last decade, it has been shown that IL4I1 is involved in the fine control of B- and T-cell adaptive immune responses. Preclinical models have revealed its role in cancer immune evasion. Recent clinical data highlight IL4I1 as a new potential prognostic marker in human melanoma. As a secreted enzyme, IL4I1 may represent an easily targetable molecule for cancer immunotherapy.

## 1. Introduction

The immune system has evolved to defend organisms from external aggressions and internal dangers, as theorized by Polly Matzinger in 1994 [1]. The most important of these mechanisms are partially redundant to limit the possibility of genetically driven escape of pathogens and tumors. Moreover, with the appearance of the adaptive arm in vertebrates, the immune system has recycled ancient mechanisms of innate defense as a means of regulating adaptive immune cell populations. In particular, biochemical alterations of the extracellular milieu by cells belonging mainly to the myeloid lineage can limit the expansion of microorganisms and negatively control the adaptive immune response. As a representative example, the enzymes that catabolize essential amino acids (aa) help to contain auxotrophic microorganisms and highly proliferating cells, such as recently activated T cells, which rely on the external uptake of essential aa to satisfy their high demand. In addition, these enzymes can produce toxic catabolites that display both anti-infectious properties and inhibitory or pro-apoptotic effects on immune cells. Thus, aa-catabolizing enzymes have acquired functions in the regulation of excessive lymphocyte activation and inflammation.

These enzymes can be classified on the basis of the aa substrate: inducible nitric oxide synthase (iNOS, the only isoform of NOS produced by immune cells) and type 1 and type 2 arginases (Arg1 and Arg2) degrade the semi-essential aa arginine; type 1 and type 2 indoleamine 2,3-dioxygenases (IDO1 and IDO2) and the tryptophan 2,3 dioxygenase (TDO) catabolize the essential aa tryptophan; and finally, the enzyme interleukin 4 (IL-4)-induced gene 1 (IL4I1) oxidizes phenylalanine. The reactions catabolized by these enzymes, as well as their principal characteristics, are described in Table 1. Apart from iNOS, all are able to fully deplete their respective aa substrate. Along with aa degradation, they also (with the exception of Arg1 and Arg2) release toxic by-products of the reaction: kynurenines for IDO and TDO, nitric oxide (NO) for iNOS, and hydrogen peroxide (H_2_O_2_), ammonia (NH_3_), and phenylpyruvate for IL4I1. Most of these enzymes are widely expressed, with Arg1 and Arg2, IDO, TDO, and iNOS being largely expressed outside of the immune system.

The enzymes catabolizing arginine and tryptophan have been the object of several exhaustive reviews. Therefore, we will focus only on the lesser-known IL4I1, which is expressed mostly by immune cells.

## 2. L-Amino-Acid Oxidase Interleukin-Four Induced Gene 1 (IL4I1)

IL4I1 was named after its mRNA was discovered in mouse B splenocytes stimulated by the cytokine interleukin 4 (IL-4) [2]. The human gene is located within the leukocyte-receptor complex on chromosome 19, whereas the mouse gene is located on chromosome 7 in a region known to be associated with susceptibility to systemic lupus erythematosus. Five isoforms of *IL4I1* are encoded by the human gene, but they should all lead to the same secreted protein, as they differ only in the 5′ untranslated region and the first two exons that encode a signal peptide. Isoform 1 is expressed in lymphoid tissue [3] and also in human spermatozoa [4]. The second isoform is expressed in rare cells of the central nervous system [5] and in human spermatozoa [4], whereas little is known about the other isoforms.

IL4I1 is a glycosylated protein that is secreted from the cells that produce it [6]. It belongs to the L-amino-acid oxidase (LAAO) family of flavin adenine dinucleotide (FAD)-bound enzymes, which are found throughout evolution, from bacteria to mammals. IL4I1 performs oxidative deamination of phenylalanine into phenylpyruvate, liberating H_2_O_2_ and NH_3_ in the process. Low activity towards tryptophan and arginine has also been described for the mouse and human enzyme, respectively [7,8]. No specific inhibitors are currently available against IL4I1. Some molecules have been shown to inhibit the related LAAOs found in snake venom, but they are generally non-selective and have little activity (molecules with broad inhibitory spectra, working in the millimolar range) [9].

## 3. Expression of IL4I1

Expression of IL4I1 has been described mainly in cells of the human immune system, with cells of myeloid origin (monocyte/macrophages and dendritic cells) showing the highest production, particularly after stimulation with inflammatory and T helper type 1 (Th1) stimuli, such as ligands for Toll-like receptors, interleukin (IL)-1β, and type I and II interferons (IFNs) [10,11]. Such induction relies on NFκB and STAT1 activation. Accordingly, IL4I1 is strongly produced by dendritic cell and macrophage populations from chronic Th1 granulomas of sarcoidosis and tuberculosis, but not Th2 granulomas (schistosomiasis). Moreover, tumor-infiltrating macrophages from various histological types of tumors strongly produce IL4I1 (see below) [12]. However, IL4I1 expression in mouse macrophages may be regulated by different mechanisms, as it was reported to be controlled by Th2 type stimuli, such as IL-4 [8].

IL4I1 is also expressed by human peripheral blood B cells stimulated by IL-4 and CD40L via the activation of the STAT6 and NF-κB pathways, although at 10-fold lower levels than by dendritic cells or macrophages [10]. IL-4 and CD40L are important signals provided by follicular T helper (TFH) cells to maturing germinal-center B cells. Thus, it is not surprising that IL4I1 is expressed in centrocytes [13,14]. Accordingly, we have detected IL4I1 in B lymphoma cells originating from germinal-center B cells, such as follicular B cell lymphoma.

Finally, IL4I1 can also be produced by certain types of T cells. Its transcriptional expression in CD4^+^ T cells is controlled by RORγT and it is thus detected in Th17 cells and T cells undergoing Th17 differentiation from naïve or regulatory T cells [15,16,17]. Recent proteomic data and our unpublished observations indicate that IL4I1 is expressed by mucosal-associated invariant T cells (MAIT), a MR1-restricted antibacterial T-cell population mostly detected at barrier sites and in the liver and blood. MAIT cells accumulate the enzyme, together with granzyme and perforin, at the immunological synapse formed with the target cell, suggesting that they secrete it during the exocytosis of cytotoxic granules [18]. IL4I1 enrichment has not been detected in conventional CD8^+^ cytotoxic T cells or NK cells. Very little is known about the murine cell populations expressing IL4I1.

In inflammatory situations where IL4I1 is strongly produced, its presence may be detectable in the plasma or serum. However, its activity is difficult to detect due to a strong background in this type of matrix and no validated ELISA is currently available.

## 4. IL4I1 Inhibition of T and B Cells

IL4I1 induces a decrease in the proliferative capacity of human T lymphocytes, associated with down-modulation of the CD3ζ chain and the diminished production of IL-2 and inflammatory cytokines and chemokines [6,10]. Our initial observations also showed that CD4^+^ and CD8^+^ T lymphocytes are equally sensitive to the action of IL4I1, whereas memory T-cell proliferation is significantly more highly affected than that of naïve T cells. However, IL4I1 is not devoid of a biological effect on naïve T cells, as it facilitates the differentiation of purified naïve CD4^+^ T cells into FoxP3^+^ regulatory T (Treg) cells [19]. A role of phenylalanine depletion and/or of the H_2_O_2_ produced by the enzymatic reaction in the modulation of T-cell proliferation and Treg cell differentiation has been demonstrated, which appears to involve modulation of the mTORC1 signaling pathway (Figure 1). In Th17 cells, the production of IL4I1 auto limits cell cycle progression [16]. The resulting proliferative blockade of this highly proinflammatory subset has been suggested to depend on the cell cycle inhibitor Tob1 [15].

More recent data show that IL4I1 is locally produced by antigen-presenting cells in the synaptic cleft formed with the T cell, where it binds to the T-lymphocyte surface in a yet undefined manner [20]. The presence of IL4I1 during T-lymphocyte activation leads to the diminution of T-cell receptor signaling at both the early (ZAP70 activation) and late (activation of the MAP kinase and NFκB pathways) steps. No effect of the products of the enzymatic reaction or phenylalanine depletion have been observed on TCR signaling, suggesting that binding to a T cell surface receptor may be involved. Indeed, biological effects independent of the enzymatic activity, including binding to extracellular receptors, have already been described for a number of enzymes, such as vascular endothelial protein (VAP) 1 and IDO1 [21,22].

The role of IL4I1 expression on B cells has been recently described using IL4I1 KO mice. Apart from the B cells, a full characterization of the immune cell populations in IL4I1 KO mice has not been published. Our personal data indicate that very few differences with WT mice are detectable at steady state, e.g., the percentages of CD4 and CD8 T cells measured in the blood, spleen, and peripheral lymph nodes are similar. This absence of overt phenotype explains why IL4I1KO mice do not suffer from any immune-related pathology in steady state, like the other mouse models deficient in an amino-acid-catabolizing enzyme (IDO1 KO, Arg1 KO, iNOS KO). IL4I1 limits BCR-induced signaling, resulting in diminished B-cell proliferation [23]. In vivo, the IL4I1 genetic deficiency is associated with an accelerated egress of immature B cells from the bone marrow, and a higher serum level of natural immunoglobulins and anti-dsDNA antibodies. Nevertheless, no clinical evidence of autoimmunity has been observed, even in 12-month-old mice. After immunization with T-dependent antigens, B-cell-derived IL4I1 limits the amplitude of the germinal center reaction and subsequent generation of terminally differentiated B cells (memory B cells and plasma cells), as well as the antibody response. In contrast, IL4I1 does not impair T-independent type 1 response.

Overall, these data suggest that IL4I1 may be involved in the control of T-cell priming and differentiation by dendritic cells, regulation of the inflammatory balance between Th17 and Treg cells, restriction of effector/memory T cell proliferation [24] and function and, finally, B cell maturation in the germinal center and BCR-dependent activation. Thus, IL4I1 may represent a checkpoint of multiple steps of the adaptive immune response. During autoimmune processes, a default of IL4I1 production may enhance the production of autoantibodies and/or the T cell response directed against autoantigens. Conversely, during the development of cancer, infiltration of the tumor microenvironment by IL4I1-secreting cells may help with tumor cell escape from the immune response. Indeed, IL4I1 can limit the proliferation and function of antitumor T cells directly or indirectly, by facilitating regulatory T cell differentiation (Figure 2). These assumptions are reinforced by in vivo data in mice and humans, as discussed below.

## 5. The Role of IL4I1 in Infection and the Control of Immunopathology

L-amino-acid oxidases represent a very ancient family of enzymes involved in the defense against infection of primitive living organisms. IL4I1 produces the well-known antiseptic H_2_O_2_ and can starve auxotrophic bacteria by degrading phenylalanine. Moreover, the bactericidal effect of H_2_O_2_ is amplified by basification induced by NH_3_, a second byproduct of the IL4I1 reaction [25]. The very recent discovery that MAITs can secrete IL4I1 following contact with a target cell suggests that IL4I1 may participate in the killing of cells infected with intracellular bacteria [18]. This is consistent with the high levels of IL4I1 detected in the macrophage-derived cells and dendritic cells of tuberculosis granulomas, where it may reduce dissemination of the pathogen while limiting excessive Th1-cell activation [10]. IL4I1 expression has also been reported in alveolar type II cells during infection by the mold *Aspergillus fumigatus* [26] and a monocytic macrophage cell line infected with *Candida albicans* [27], although its role in the antifungal response was not established. Finally, IL4I1 induction has been reported in viral infections with influenza virus, human immunodeficiency virus, and avian leukosis virus [28,29,30].

Thus, IL4I1 is induced by infection with various types of pathogens and may contribute to both the containment of the infectious agent and the limitation of the immunopathology mediated by effector Th1 or Th17 cells. Defects in the latter function may facilitate the development of autoimmunity, as observed in murine models. Indeed, IL4I1 mRNA decreases in MRL/lpr mice over the course of a disease that resembles human systemic lupus erythematosus [31]. The expression of IL4I1 by macrophages increases M2 polarization in an autocrine manner in a model of experimental autoimmune encephalitis [8] and promotes the arrest of inflammation and re-myelinization of the central nervous system [32].

## 6. The Role of IL4I1 in the Immune Escape of Tumors

The mRNA of IL4I1 was initially found to be overexpressed in a rare subtype of aggressive human B-cell lymphoma, primary mediastinal B-cell lymphoma (PMBL) [33]. This observation led us to conduct a large immunochemistry study on more than 300 human tumor biopsies, representing 30 different histological types of lymphomas and non-hematological cancers. This study identified IL4I1 as a marker of macrophage infiltration, as the oxidase was expressed by tumor-infiltrating macrophages in almost all tumors analyzed. Moreover, IL4I1 expression was observable in tumor cells in specific types of cancer, mainly mesothelioma and B cell lymphomas derived from germinal-center B cells, such as PMBL and follicular lymphoma [12]. In follicular lymphoma, the presence of IL4I1 probably results from the persistence of its physiological expression in centrocytes and may thus contribute to limiting proliferation of the tumor cells. This may explain why IL4I1 expression in patients with follicular lymphoma is associated with parameters indicative of a good prognosis [12]. For the moment, no studies correlating IL4I1 expression with that of the other immunosuppressive enzymes have been done in human cancers. This type of study could be important to give a more precise picture of the immunosuppressive landscape of tumors, especially since some of these enzymes share inducing stimuli.

On the contrary, in patients with non-hematological tumors infiltrated by macrophages, IL4I1 is expected to negatively affect the evolution of the cancer due to its suppressive effect on the T-cell response. This hypothesis is supported by studies performed in mouse models of melanoma (Figure 2).

In a model of the adoptive transfer of tumor cells expressing the enzyme or not, the incidence of tumor development was enhanced and the CD8^+^ T-cell response against tumor antigens was diminished by IL4I1 [34]. In a second mouse model of spontaneous melanoma development, the genetic absence of IL4I1 delayed the appearance of primary tumors and metastases and favored tumor infiltration by T and B cells, while diminishing the infiltration by polymorphonuclear myeloid-derived suppressor cells [34,35]. In human primary cutaneous melanoma, the infiltration by IL4I1^+^ cells correlated positively with the presence of Treg cells and negatively with the presence of functional cytotoxic T cells. In a recent review, we addressed emerging questions about the impact of IL4I1 on B-cell functions in melanoma [36]. Finally, we showed a clinical correlation between IL4I1 expression in primary cutaneous tumors and parameters of poor outcome in patients with melanoma. It was also associated with invasion of the sentinel lymph node, a higher melanoma stage, and rapid relapse and tended to correlate with shorter overall survival, specifically when considering cases in which the IL4I1^+^ cells were in close contact with tumor cells [37]. IL4I1 has also been suggested to be a significant prognostic parameter in breast cancer, renal cancer, and glioma [38,39] (www.proteinatlas.org). Profound metabolic changes have been found in renal cell carcinoma, which included a strong increase in IL4I1 [40]. These data all point to a key role of IL4I1 in the immune evasion of tumors.

## 7. Conclusions

IL4I1 is a phenylalanine oxidase with anti-infectious and immunoregulatory properties. Although little is known about its mechanisms of action, we are starting to understand that IL4I1 plays a significant role in the biology of B cells and in the control of the cross-talk between antigen-presenting cells and T cells in a vast variety of situations. Manipulating IL4I1 in vivo thus opens new avenues for the treatment of autoimmunity and cancer. In particular, the chemical blockade of IL4I1 activity in patients with cancer may contribute to the restoration of specific anti-tumor immune responses.

## Figures and Tables

**Figure 1 cells-08-00757-f001:**
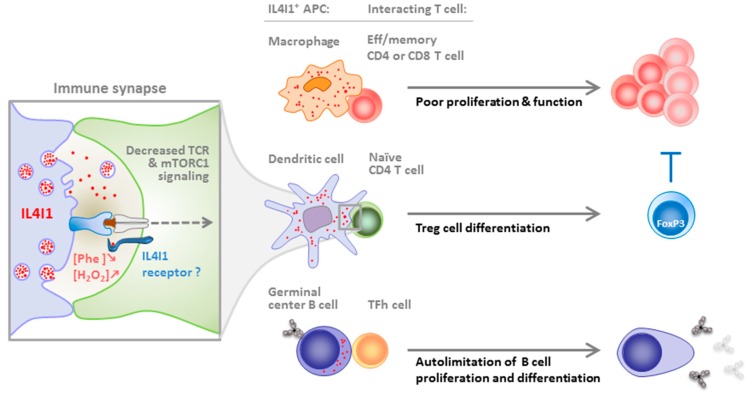
Regulation of the immune response by IL4I1. IL4I1 is produced by antigen-presenting cells (APC) when they are activated with appropriate stimuli. It is secreted at the immune synapse formed with T cells and possibly binds to a yet unknown receptor that might concentrate the enzyme locally or transmit an intracellular signal to the T cell (inset). Indeed, IL4I1 downmodulates the signaling pathways downstream of the TCR independently of its enzymatic activity. IL4I1 also diminishes mTORC1 signaling, probably due to H_2_O_2_ production and/or phenylalanine depletion. The consequence of the APC-T cell crosstalk depends on the nature of the two partners. Effector and memory CD8^+^ and CD4^+^ T cells are sensitive to IL4I1 inhibition of their proliferation and effector functions; the differentiation of naïve CD4^+^ T cells is biased towards FoxP3^+^ regulatory T (Treg) cells (Treg cells can also contribute to the suppression of effector T cells); and B cells in the germinal center, which upregulate IL4I1 expression in response to stimuli from the follicular T helper cell, are restrained in their proliferation and terminal differentiation in memory cells and antibody-secreting plasmocytes. The effect of IL4I1 on the TFh cell has not been explored.

**Figure 2 cells-08-00757-f002:**
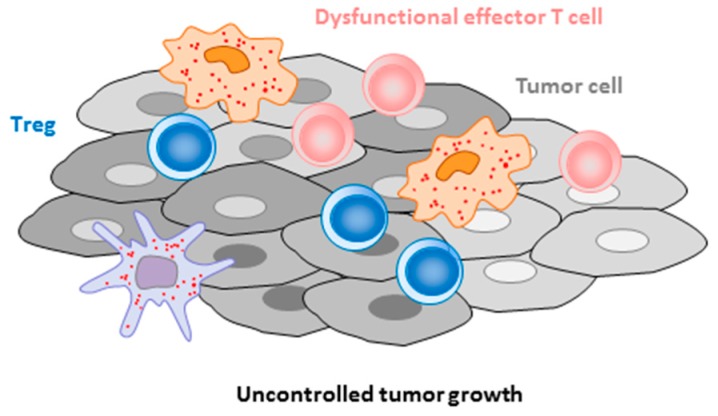
Role of IL4I1 in cancer immune escape. The presence of IL4I1-producing cells in the tumor cell microenvironment restrains the anti-tumor immune response by directly limiting the proliferation and functionality of cytotoxic T cells and Th1 cells, or indirectly by facilitating the accumulation of Treg cells. In murine models, the level of IL4I1 expression is associated with tumor growth. In melanoma patients, the presence of IL4I1-expressing cells in close contact with tumor cells correlates with parameters of poor outcome.

**Table 1 cells-08-00757-t001:** Characteristics of the best-known amino-acid-catabolizing enzymes.

Enzyme Acronym	Inducible Nitric Oxide Synthase (iNOS)	Arginase (Arg1 and Arg2)	Indoleamine 2,3-Dioxygenase (IDO1 and IDO2)	Tryptophan 2,3 Dioxygenase (TDO)	Interleukin 4-Induced Gene 1 (IL4I1)
**Substrate and reaction**	Arginine→ Citrulline + NO	Arginine→ Ornithine + Urea	Tryptophan +O_2_→ Kynurenines	Tryptophan +O_2_→ Kynurenines	Phe→ H_2_O_2_ + NH_3_ + Phenylpyruvate
**Expression in the immune system**	Myeloid cells	Myeloid cells	Myeloid cells	Myeloid cells	Myeloid cells, T and B lymphocytes
**Control by soluble factors**	IFNγ	IL-4 and IL-10 in mouse macrophages	IFNγ	Corticosteroids and glucagon	Type I and II IFN in myeloid cells (humans) and IL-4 in B cells (humans and mice)
**Localization**	Intracellular	Intracellular	Intracellular	Intracellular	Secreted

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
