# Peer review of "The IL4I1 Enzyme: A New Player in the Immunosuppressive Tumor Microenvironment"

_cells, 2019, doi:10.3390/cells8070757_

Round 1
Reviewer 1 Report
This manuscript gives an overview of the immunologic functions of IL4I1 and its implication in the tumor microenvironment. The manuscript is comprehensive and nicely written.
Major comments :
- The authors should discuss more in depth the phenotype of IL4I1 knockout mice. Apparently, these mice display defects in B cell differentiation and function, but nothing is mentioned about frequencies and functions of other immune cell types in these mice.
- The authors should reference : Romagnani S. (Eur J Immunol. 2016 Oct;46(10):2302-2305), who nicely discuss the effect of IL4I1 on T cells.
- IL4I1 seems to be mainly involved in B cell malignancies and maybe melanoma, but much less in other solid tumor types as described in reference 10. The number of samples studied per tumor type in that study are also very limited, which makes it difficult to draw firm conclusions. Would it not be needed to perform larger studies on IL4I1 expression across tumor types wherein also correlation with other immunosuppressive markers is assessed ?
- Is IL4I1 and/or its activity detectable in serum or plasma ?
- In the conclusion, the authors mention IL4I1 blockade as a novel treatment option for autoimmunity and cancer. Are there blocking antibodies or inhibitors available ? To my knowledge, no mouse studies on IL4I1 blockade have been described ?
Minor comments :
- Line 48 : immune-cell should be immune cell
- Line 52 : recently-activated should be recently activated
- Line 90 : FAD-bound enzymes : write FAD in full the first time it is used
- Line 112 : certain types of T cell should be certain types of T cells
- Line 131 : mTOR-C1 should be mTORC1
- Line 132 : cell-cycle should be cell cycle
- Line 151 : by antigen-presenting cell should be by antigen-presenting cells
- Line 182 : This consistent with should be This is consistent with
- Line 188 : influenzas virus should be influenza virus
- Line 201 : reference is not formatted
Author Response
Reviewer 2
This is a very good review of IL4I1 which is gaining importance as a tumor target.
IL4I1 is a Lysosomal L-amino-acid oxidase with highest specific activity with phenylalanine along with few other amino acids with limited activity. As name indicates, its expression is induced by IL-4 and its catalytic activity is accompanied by binding to flavin adenine dinucleotide cofactor. It is expressed in various immune cells and its main function is anti-inflammatory by suppressing over-expression of T cells. Tumors known to exploit anti-inflammatory pathways to suppress immune functions for their growth benefit and evade immune system. IL4I1 is also targeted in similar fashion by tumors to escape from immune system.
Authors have nicely described its expression and functions. It would easy understanding of readers if authors describe with more clarity on its functions as anti-inflammatory, the reason why it inhibits proliferation of immune cells. This is exact same reason why tumors exploit these anti-inflammatory pathways so as to escape the immune recognition.
We have taken in consideration the reviewer’s suggestion and added few sentences at line 187-192.

Reviewer 2 Report
This is a very good review of IL4I1 which is gaining importance as a tumor target.
IL4I1 is a Lysosomal L-amino-acid oxidase with highest specific activity with phenylalanine along with few other amino acids with limited activity. As name indicates, its expression is induced by IL-4 and its catalytic activity is accompanied by binding to flavin adenine dinucleotide cofactor. It is expressed in various immune cells and its main function is anti-inflammatory by suppressing over-expression of T cells. Tumors known to exploit anti-inflammatory pathways to suppress immune functions for their growth benefit and evade immune system. IL4I1 is also targeted in similar fashion by tumors to escape from immune system.
Authors have nicely described its expression and functions. It would easy understanding of readers if authors describe with more clarity on its functions as anti-inflammatory, the reason why it inhibits proliferation of immune cells. This is exact same reason why tumors exploit these anti-inflammatory pathways so as to escape the immune recognition.
Author Response
Response to the reviewers’ comments
Reviewer 1
This manuscript gives an overview of the immunologic functions of IL4I1 and its implication in the tumor microenvironment. The manuscript is comprehensive and nicely written.
Major comments :
- The authors should discuss more in depth the phenotype of IL4I1 knockout mice. Apparently, these mice display defects in B cell differentiation and function, but nothing is mentioned about frequencies and functions of other immune cell types in these mice.
We have added some sentences regarding this question at the end of the paragraph, which explains the modifications of B cell populations in IL4I1 KO mice (from line 168 to line 174).
- The authors should reference : Romagnani S. (Eur J Immunol. 2016 Oct;46(10):2302-2305), who nicely discuss the effect of IL4I1 on T cells.
We have added this reference line 185 (reference 24).
- IL4I1 seems to be mainly involved in B cell malignancies and maybe melanoma, but much less in other solid tumor types as described in reference 10. The number of samples studied per tumor type in that study are also very limited, which makes it difficult to draw firm conclusions. Would it not be needed to perform larger studies on IL4I1 expression across tumor types wherein also correlation with other immunosuppressive markers is assessed ?
A 315 sample study is not so small but it is true that no studies have been published looking at IL4I1 together with other immunosuppressive enzymes. We have added two sentences indicating this interesting aspect at lines 230-233.
- Is IL4I1 and/or its activity detectable in serum or plasma ?
We have added a few sentences regarding this interesting question at the end of the paragraph “Expression of IL4I1”, line 126-128.
- In the conclusion, the authors mention IL4I1 blockade as a novel treatment option for autoimmunity and cancer. Are there blocking antibodies or inhibitors available ? To my knowledge, no mouse studies on IL4I1 blockade have been described ?
None of the available anti-IL4I1 antibodies have been described to display a blocking activity. Our group has recently developed potent rabbit monoclonal antibodies, which are very sensitive in various immunological techniques, from immunohistochemistry to Western blot and immunoprecipitation. Some commercial antibodies are also available but they often do not have been properly validated. As it concerns the inhibitors, few compounds have been reported, with little specificity (in particular, it is not clear if they are able to inhibit the peroxidase used in the enzymatic test to determine IL4I1 activity).
We have added a sentence about this aspect at the end of paragraph “L-amino-acid oxidase Interleukin-four induced gene 1 (IL4I1)” lines 94-97.
Minor comments :
- Line 48 : immune-cell should be immune cell
- Line 52 : recently-activated should be recently activated
- Line 90 : FAD-bound enzymes : write FAD in full the first time it is used
- Line 112 : certain types of T cell should be certain types of T cells
- Line 131 : mTOR-C1 should be mTORC1
- Line 132 : cell-cycle should be cell cycle
- Line 151 : by antigen-presenting cell should be by antigen-presenting cells
- Line 182 : This consistent with should be This is consistent with
- Line 188 : influenzas virus should be influenza virus
- Line 201 : reference is not formatted
Some of the suggested corrections (previously line 48, 52, 132 and 151) are compound adjectives that modify the following noun; thus they need a hyphen as specified by our English speaker corrector (Alex Edelman & Associates). Though, if the reviewer really insists we can remove them. All the remaining corrections have been done (the modified text is in red).